# Assessment of Spatiotemporal Variability of Evapotranspiration and Its Governing Factors in a Mountainous Watershed

**Anh Phuong Tran [1,2,\*], Joseph Rungee [3] , Boris Faybishenko [1], Baptiste Dafflon [1] and Susan S. Hubbard [1]**

[1]  Earth and Environmental Sciences Area, Lawrence Berkeley National Laboratory, Berkeley, CA 94720, USA; bafaybishenko@lbl.gov (B.F.); BDafflon@lbl.gov (B.D.); sshubbard@lbl.gov (S.S.H.)
[2]  Water Research Institute, Lang Thuong, Dong Da, Hanoi 11512, Vietnam
[3]  Sierra Nevada Research Institute, University of California, Merced, CA 95343, USA; jrungee@ucmerced.edu
[\*]  Correspondence: phuongtran.monre@gmail.com; Tel.: 84-24-62811218

**Abstract:** Evapotranspiration (ET) is a key component of the water balance, which influences hydrometeorology, water resources, carbon and other biogeochemical cycles, and ecosystem diversity. This study aims to investigate the spatio-temporal variations of ET at the East River watershed in Colorado and analyze the factors that control these variations. ET was acquired using the community land model (CLM) simulations and was compared with the values estimated using Fu's equation and a watershed-scale water balance equation. The simulation results showed that 55% of annual precipitation in the East River is lost to ET, in which 75% of the ET comes from the summer months (May to September). We also found that the contribution of transpiration to the total ET was ~50%, which is much larger than that of soil evaporation (32%) and canopy evaporation (18%). Spatial analysis indicated that the ET is greater at elevations of 2950–3200 m and lower along the river valley (<2750 m) and at the high elevations (>3900 m). A correlation analysis of factors affecting ET showed that the land elevation, air temperature, and vegetation are closely correlated and together they govern the ET spatial variability. The results also suggested that ET in areas with more finely textured soil is slightly larger than regions with coarse-texture soil. This study presents a promising approach to the assessment of ET with a high spatiotemporal resolution over watershed scales and investigates factors controlling ET spatiotemporal variations.

**Keywords:** evapotranspiration; spatio-temporal variations; community land model; topography; vegetation; air temperature; soil texture; PRISM; daymet

## 1. Introduction

Evapotranspiration (ET) is an important component of water and energy balance at the land surface. ET is crucial for hydrometeorological prediction and influences carbon cycling, water availability, and ecosystem diversity. At the watershed scale, information about ET is particularly valuable for separating precipitation into infiltration and runoff, which is important for estimating downgradient water resources, as well as for investigating hydrology-driven biogeochemical dynamics. However, ET remains the most difficult component of the land-surface water budget to estimate, primarily due to an inability to directly measure ET and the complex vegetation-soil-atmosphere interactions that influence it.

Many efforts have been made to estimate ET using various multi-scale observations. For example, it can be estimated using local techniques such as weighing lysimeters [1], the Bowen ratio-energy balance [2,3] and plant chamber [4]. ET can also be estimated using semi-analytical formulae based

on a combination of meteorological data and empirical parameters [5–9]. The third key category of approaches that can be used to estimate ET is the use of process-based land surface models to simulate the physical processes of heat and water in the vegetation-soil-bedrock continuum. Well-known land surface models include the Community Land Model (CLM) [10], Variable Infiltration Capacity model (VIC) [11], CoupModel [12] and Noah [13]. Compared to the analytical approach, the land surface modeling approach accounts for more of the processes that influence ET, including the exchanges of energy, heat and water from the top of the canopy to the bedrock. This study explores the reliability of the land surface model, CLM in ET estimation and compares the results with those obtained by two semi-analytical methods, namely a modified Budyko's curve (Fu's equation), and the watershed-scale water balance equation.

Investigation into the factors that control ET such as topography, vegetation, soil texture and meteorological forcing was also conducted. For example, Peel et al. [14] analyzed data from 699 catchments over the world and found that the mean annual ET in tropical and temperate forested catchments is much higher than in non-forested catchments. By contrast, in cold regions, evapotranspiration in forested catchments is considerably lower than that in non-forested catchments. Nepstad et al. [15] and Hodnett et al. [16] indicated that during dry periods, the ET from forested regions is larger than that in grasslands because the forest trees have deeper roots than grass, which enable them to take water from deep soil layers. Further, Detto et al. [17] showed that the actual transpiration of woody vegetation is close to the potential transpiration even in the driest conditions, while the actual transpiration of grass is strongly influenced by surface soil moisture availability. McVicar et al. [18] considered the influence of topography on ET via the dependence of the maximum and minimum atmospheric temperature, wind speed, vapor pressure and radiation on the elevation, slope and aspect. Bennie et al. [19] considered the effects of slope and aspect of a vegetated surface on ET by reference to their impacts on the solar radiation. By using a distributed hydrological model, Chen et al. [20] showed that topography strongly influenced ET, soil moisture and groundwater table. Luxmoore and Sharma [21] indicated that annual ET was higher in fine textured soils than in coarse textured soils. Kollet [22] showed that soil texture strongly influenced evapotranspiration in dry conditions but had a negligible effect in wet conditions. While these studies document various factors influencing ET, no study has comprehensively considered the impacts of all of these factors on ET at the watershed scale in mountainous regions.

The objective of this study therefore is twofold: (1) Develop an approach to estimating the actual ET with a high spatial and temporal resolution and assess the spatiotemporal variability over a mountainous Upper East River, CO watershed, and (2) Analyze the relationship between ET and the factors that potentially influence its spatiotemporal variability. To accomplish this, CLM simulations were performed to estimate ET at the East River watershed. Then, the obtained results wre compared with the annual evapotranspiration calculated using two different semi-analytical methods: a modified Budyko's curve (Fu's equation) and the watershed-scale water balance equation. Next, we investigated the spatiotemporal variations of ET over annual and inter-annual durations and quantified the contribution of soil evaporation, canopy evaporation and transpiration to the total ET. Finally, the dependence of ET on factors such as meteorological forcing variables, elevation, vegetation and soil properties over space and time are explored.

## 2. Methodology and Materials

### 2.1. Models for ET Calculation

In this study, we considered three methods of calculating ET, namely the numerical CLM model, the water balance-based estimation, and the analytical Fu equation model (Fu, 1981). While CLM calculates the daily ET at every grid cell (with a size of $900 \times 900$ m$^2$) over the watershed region, the analytical methods provide a simple estimate of the annual ET over the watershed scale. Although the methods make different assumptions and involve different datasets, their comparison is important

for validation and gaining confidence in the CLM-based ET estimation, which will be used to explore spatiotemporal signatures and controls.

### 2.1.1. CLM Model

CLM is a physically-based model that simulates the hydro-thermal dynamics from the top of the canopy to the bedrock. CLM can simulate multiple processes at the land surface such as evaporation, transpiration, snow melt, snow accumulation and plant water uptake. In the subsurface, water and heat dynamics are simulated in a vertical soil column. In this study, we discretized the soil columns into 32 layers, which include the top 27 soil layers with a total thickness of 1.5 m, and 5 bottom bedrock layers with a total thickness of 2 m. While water dynamics are simulated using Richards' equation in 27 soil layers, heat transport is simulated using the diffusion equation in both the 27 soil layers and 5 bedrock layers. Importantly for ET calculation, the CLM separately calculates soil evaporation from the soil, canopy evaporation from the wet canopy, and transpiration from the dry canopy based on the mass and energy balance equations and Monin-Obukhov Similarity Theory. Oleson et al. [10] provided a more detailed description of the procedure used to calculate ET.

As for the CLM input data, meteorological data required by this model include precipitation, air temperature, solar radiation, wind speed, humidity and barometric pressure. The land surface characteristic inputs are topographical slope and vegetation. These inputs control the partitioning of total water into runoff, evapotranspiration and infiltration at the land surface. The final inputs required by the model are the sand, clay and organic carbon content in soil. These properties are used by the model to calculate the soil hydrological (water retention curve and saturated hydraulic conductivity) and thermal parameters (thermal conductivity and heat capacity). In Section 3.4, we evaluate the influences of these inputs on the spatiotemporal variations of ET.

### 2.1.2. Fu's Equation

In this study, we used the presentation of the Budyko model using a 1-parameter Fu's equation [23] given by:

$$\frac{ET}{P} = 1 + \frac{ET_0}{P} - \left(1 + \left(\frac{ET_0}{P}\right)^m\right)^{\frac{1}{m}} \tag{1}$$

in which $P$ is precipitation (mm); $ET_0$ is the potential ET (mm); $m$ is an empirical coefficient ranging from 1 to $+\infty$, but which usually varies in the range from 1 to 4 [7]. In this study, we estimated this coefficient by comparing ET estimated by this method with that estimated by the watershed-scale water balance equation.

Monthly $ET_0$ is calculated from the air temperature based on Hamon's equation as follows:

$$ET_0 = \begin{cases} k \times 0.165 \times 216.7 \times N \times \frac{e_s}{T+273.3} & if \quad T > 0 \\ 0 & if \quad T \leq 0 \end{cases} \tag{2}$$

in which $k$ is a dimensionless coefficient; $T$ is the average monthly air temperature (°C); and $e_s$ is the saturated vapor pressure (Pa), which is calculated from the air temperature as:

$$e_s = 6.108 \frac{17.27}{T + 237.3} \tag{3}$$

In Equation (4), $N$ is the daytime length in units of 12 h and calculated as:

$$N = \frac{24}{\pi} \omega \tag{4}$$

in which $\omega$ is the sunset angle in radians, which is related to the latitude of the study location ($\varphi$) and the Julian day of year (DOY):

$$\omega = \cos^{-1}\left\{-\tan\left[1 + 0.033\cos\left(\frac{2\pi}{365}DOY\right)\right]\tan(\varphi)\right\} \tag{5}$$

It is worth noting that in the East River, snow that accumulates during the winter months can contribute to evapotranspiration in summer months when snow melts. As a result, we could not estimate monthly ET using Fu's equation for this case study. The monthly $ET_0$ values calculated by Equation (2) are aggregated on an annual basis to use in the ET calculation in Equation (1).

### 2.1.3. Watershed-Scale Water Balance Equation

At the watershed scale, the soil water balance equation is written as:

$$\frac{dS}{dt} = P - Q - ET \tag{6}$$

If we apply the water balance Equation (12) over the water-year time scale, the change in the soil water storage change can be assumed to be negligible (i.e., $dS/dt \approx 0$) (how this assumption influences the ET estimate will be discussed in Section 3.1), which simplifies Equation (6) to:

$$ET = P - Q \tag{7}$$

in which $P$ is the average precipitation over the watershed (mm/year) and $Q$ is the river discharge at the outlet of the watershed (mm/year). It is worth noting that this water balance equation assumed that surface and groundwater have a common contributing area. Similar to the Fu method, this method is only used for calculation of annual the ET. Because the winter season begins from October, Equation (7) is only valid for the water year from October to September. Consequently, we will use this water year as a basis for calculating annual ET.

### 2.2. Data Availability and Processing

### 2.2.1. Study Site

Our study area is the Upper East River Watershed, a headwaters catchment located in the Upper Colorado River Basin in Colorado (Figure 1a), which is the focus of the Department of Energy Watershed Function Project. We delineated the watershed using the ArcGIS watershed tool with NASA's Shuttle Radar Topography Mission 30-m digital elevation model (DEM) data and coordinates for the United States Geological Survey's (USGS) stream gauge No. 09112200 as the watershed outlet (Figure 1b). The total area of this watershed analysis is approximately 64 km². We divided the whole watershed into 788 grid cells, each with a size of $900 \times 900$ m² and performed CLM simulations at each grid cell to calculate ET. The simulation period is 22 years, ranging from 1 January 1993 to 31 December 2014 with a daily time step. Below we describe the different watershed characteristics, soil properties and meteorological forcing data, which were used to estimate ET using different methods.

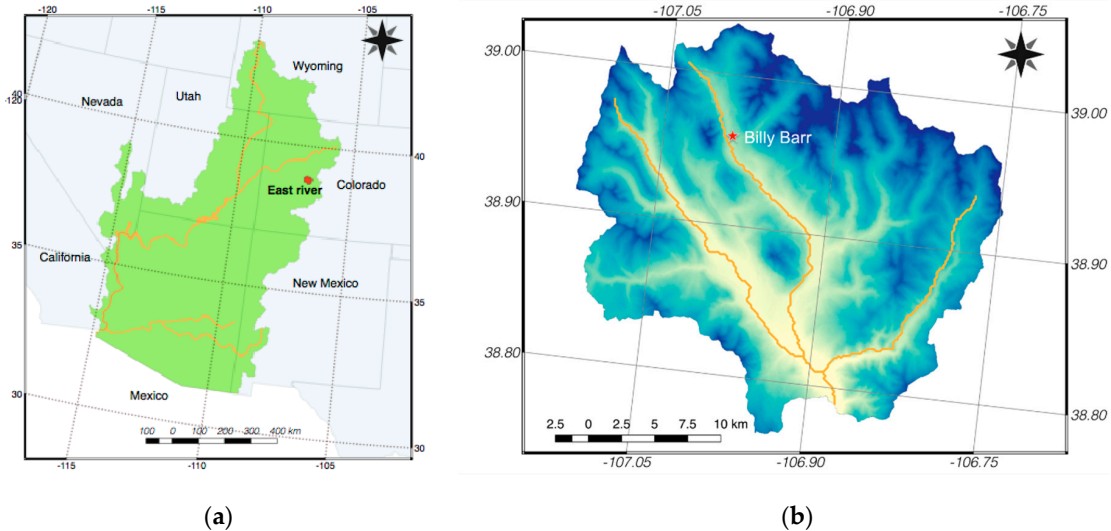

(**a**)                                                   (**b**)

**Figure 1.** (**a**) The location of the Upper East River, CO Watershed study site in the Upper Colorado River Basin (red region). (**b**) Topography of the Upper East River, CO Watershed and associated drainage system, and the location of the Billy Barr meteorological station, which provided wind speed input for the evapotranspiration (ET) analysis.

### 2.2.2. Meteorological Forcing Data

Meteorological forcing inputs for the CLM simulation included precipitation, air temperature, shortwave radiation, wind speed and humidity. To simulate ET using CLM, we used the gridded daily maps of meteorological forcing data as inputs for the CLM simulations. The gridded daily precipitation and mean air temperature data with a spatial resolution of $800 \times 800$ m$^2$ were obtained from the *PRISM* database (Daly et al. [24]). The $1000 \times 1000$ m$^2$ gridded data of shortwave radiation and vapor pressure were obtained from the *Daymet* database, version 3 (Thornton et al., [25]). The relative humidity was calculated from the air temperature and vapor pressure as:

$$e_s = 6.112 \frac{e^{17.67 \times T}}{T + 243.5} \times 100 \tag{8}$$

$$RH = \frac{e}{e_s} \times 100 \tag{9}$$

where $e_s$ is the saturated water vapor pressure (Pa); $e$ is the actual water vapor pressure (Pa) at air temperature $T$ (°C) and $RH$ is the relative humidity (%). All of these data were interpolated to the simulation map ($900 \times 900$ m$^2$ DEM map in Figure 3a). It is worth noting that both *PRISM* and *Daymet* databases considered the impact of topographical factors (e.g., aspect, slope and elevation) on the spatial distribution of meteorological forcing data. Examples of atmospheric temperature, precipitation, shortwave radiation and relative humidity on 8 August 1994 are shown in Figure 5. A comparison of Figures 2 and 3a shows that there is a close correlation between the elevation and the meteorological data. For example, the precipitation, humidity and shortwave radiation gradually increase with increasing elevation. Details on these dependences are discussed in Section 3.4.

Given both *PRISM* and *Daymet* databases do not provide wind speed data, we used measured wind speed data from the Billy Barr meteorological station (see Figure 1 for location) to represent the entire watershed and simulation period, which may influence the accuracy of the CLM simulation. Due to a lack of spatial information on wind speed, we could not evaluate the impact of this variable

on ET. In addition, due to a lack of atmospheric pressure data ($P$ (Pa)), we calculated it from the topographical elevation ($h$ (m)) as:

$$P = 101300 \left( \frac{293 - 0.0065h}{293} \right)^{5.26} \tag{10}$$

This formula allows for consideration of the effect of topographical elevation on atmospheric pressure, i.e., atmospheric pressure is lower for higher elevation. At each location, the atmospheric pressure will temporally be a constant value over the simulation period.

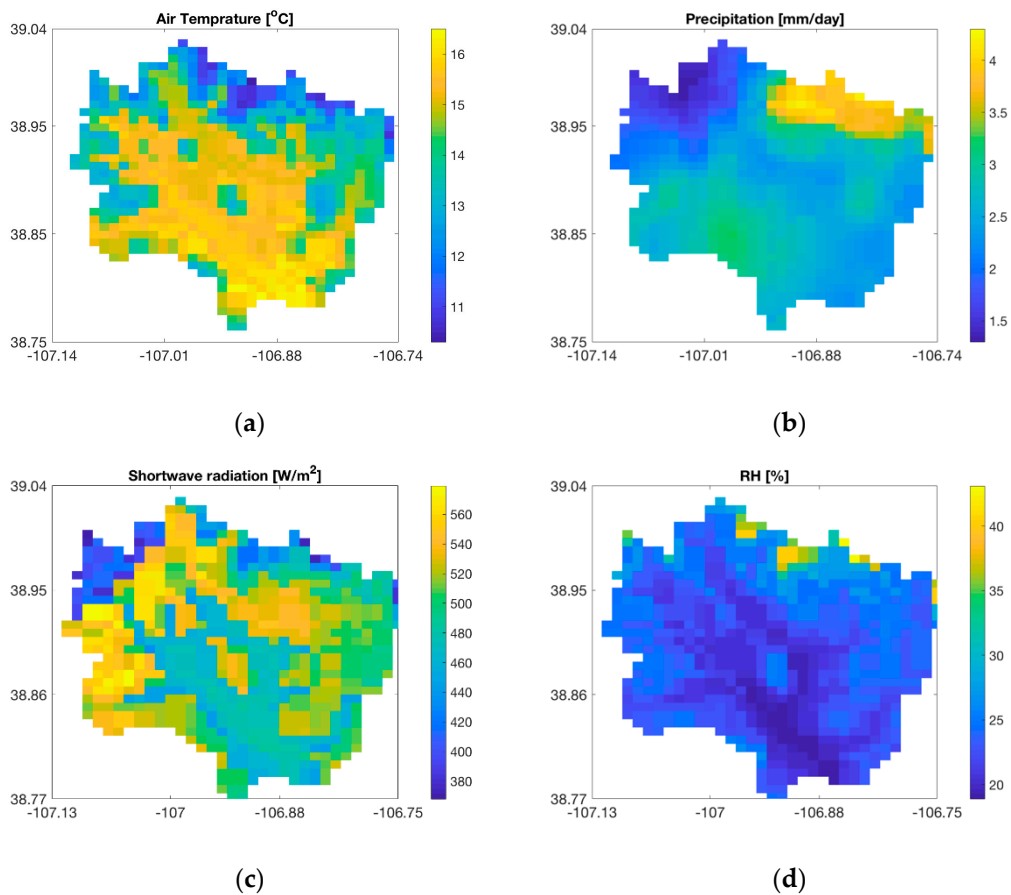

**Figure 2.** Examples of gridded air temperature (**a**), precipitation (**b**), shortwave radiation (**c**) and relative humidity (**d**) on 8 August 1994 that were used for community land model (CLM) simulation to calculate ET.

### 2.2.3. Land Surface and Soil Property Datasets

*Topographical Datasets.*

Because the precipitation and air temperature datasets have a spatial resolution of $800 \times 800$ m$^2$ and the spatial resolution of shortwave radiation and atmospheric humidity data is $1000 \times 1000$ m$^2$, we rescaled the original $30 \times 30$ m$^2$ digital elevation model (DEM) to $900 \times 900$ m$^2$ DEM and estimated ET at this scale. We created the topographical slope map based on this rescaled $900 \times 900$ m$^2$ DEM using the algorithm proposed by Burrough and McDonell [26]. Whilst slope is one of the CLM inputs, DEM was used to evaluate its indirect effect on ET via other factors such as meteorological forcing variables and vegetation. Figure 3 indicates that the watershed topography ranges from 2600 to 4300 m, with the highest elevation located at the northeast of the study site. The slope of the watershed is relatively high, ranging from 0.4° to 22°.

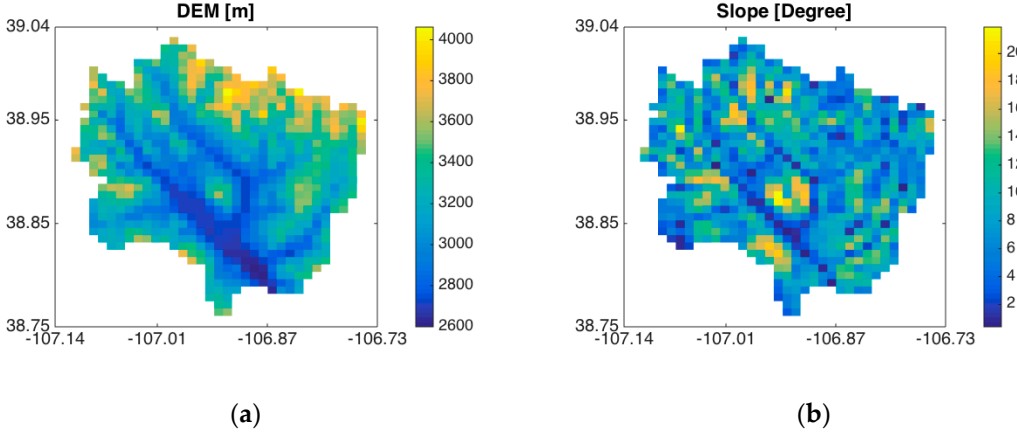

(**a**)      (**b**)

**Figure 3.** The rescaled 900 × 900 m² maps of (**a**) DEM and (**b**) topographical slope.

*Vegetation Datasets*

A vegetation map for the watershed was extracted from the 2006 National Land Cover Dataset (NLCD, USGS) with a spatial resolution of 30 m [27]. Conversion from the USGS vegetation types to CLM plant functional type was performed using the conversion method proposed by Ke et al. [28]. Figure 4 shows the plant functional type map that we used in this study. The figure indicates that there are six major plant functional types in the study site, including 49% needle evergreen—temperate; 6% needle evergreen—temperate boreal; 2% broad evergreen shrubs temperate; 30% C3 Arctic; and 2% C4 grass. In the site, 10% is bare soil, with a predominance in the upper elevations. We aggregated this map to determine the percentage of each plant functional type in 900-m spatial resolution map. CLM has a library that contains twelve values for twelve months (January to December) of LAI and plant top and bottom heights for each plant functional type, which we used to evaluate the influence of both spatial and temporal variation of vegetation on ET.

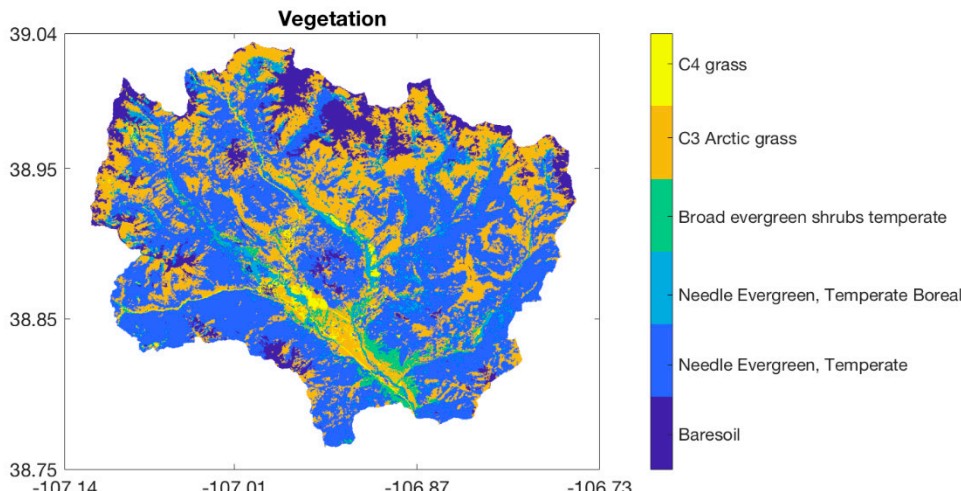

**Figure 4.** CLM plant functional type map converted from the 2006 National Land Cover Dataset (NLCD) map.

*Soil Properties*

CLM requires soil organic carbon, sand, and clay content to estimate the soil's physical parameters (e.g., hydraulic conductivity, retention curve parameters, thermal conductivity and heat capacity). In this study, we used a soils dataset obtained from the Natural Resources Conservation Service of the U.S. Department of Agriculture (NRCS) to derive these soil properties. Figure 5 illustrates the estimated organic carbon, sand and clay content over the East River Watershed. As shown in the figures, sand

constitutes the majority of the soil mixture (35–66%), followed by clay (9–30%), then organic carbon (<4%).

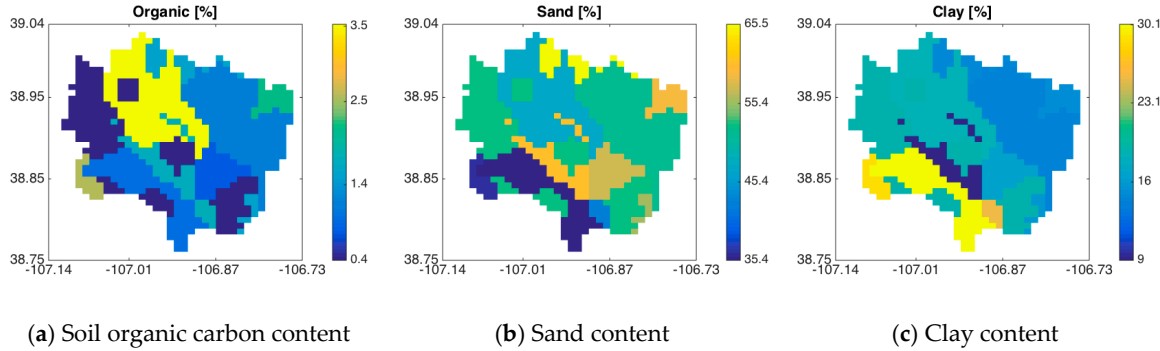

(**a**) Soil organic carbon content  (**b**) Sand content  (**c**) Clay content

**Figure 5.** Maps of (**a**) soil organic carbon, (**b**) sand and (**c**) clay content estimated using the Natural Resources Conservation Service (NRCS) soil database.

## 3. Results

### 3.1. Evapotranspiration Estimation

Figure 6a compares the ET estimated by the CLM model (annually-averaged for the water-year October–September of each year) with the values obtained using Fu's equation and the watershed-based water balance equation. The annual ET values obtained from the CLM simulations at different grid cells were spatially averaged over the whole watershed for this comparison. For Budyko's model, we adjusted parameter *m* in Equation (1). The value of *m* that provides the best agreement between Fu's equation calculation and CLM simulation and water-balance ET is *m* = 2.1. The agreement between the three approaches is determined by criteria bias and root mean square error (RMSE) as below:

$$bias = \frac{\sum_{i=1}^{N} ET_{cal}^i}{\sum_{i=1}^{N} ET_{ref}^i} \tag{11}$$

$$RMSE = \sqrt{\frac{1}{N} \sum_{i=1}^{N} \left( ET_{cal}^i - ET_{ref}^i \right)^2} \tag{12}$$

in which $ET_{cal}^i$ and $ET_{ref}^i$ are respectively, the calibration ET (CLM/Fu ET) and water-balance ET in year $i^{th}$; *N* is the number of years (*N* = 21). Figure 6a indicates that there is relatively good agreement among the three methods. The *RMSE* is, respectively, 49.3 and 50.7 (mm) for the *ET* calculated by the CLM model and Fu's equation. The bias criterion for both CLM simulation and Fu's equation in comparison with the water-balance ET is equal to 1. However, when the annual ET is less than 500 mm, ET estimated by the CLM model and Fu's equation is larger than by the watershed-scale water balance equation. By contrast, when the annual ET is smaller, ET obtained by the watershed-scale water balance equation is larger. This relationship stems from the assumption that the soil water storage does not change over time ($\frac{dS}{dt} = 0$) in the watershed-scale water balance. In fact, in low precipitation years, soil is drier and $\frac{dS}{dt} < 0$ and in high precipitation year, soil is wetter and $\frac{dS}{dt} > 0$. As a result, ET is larger in dry years and smaller in wet years.

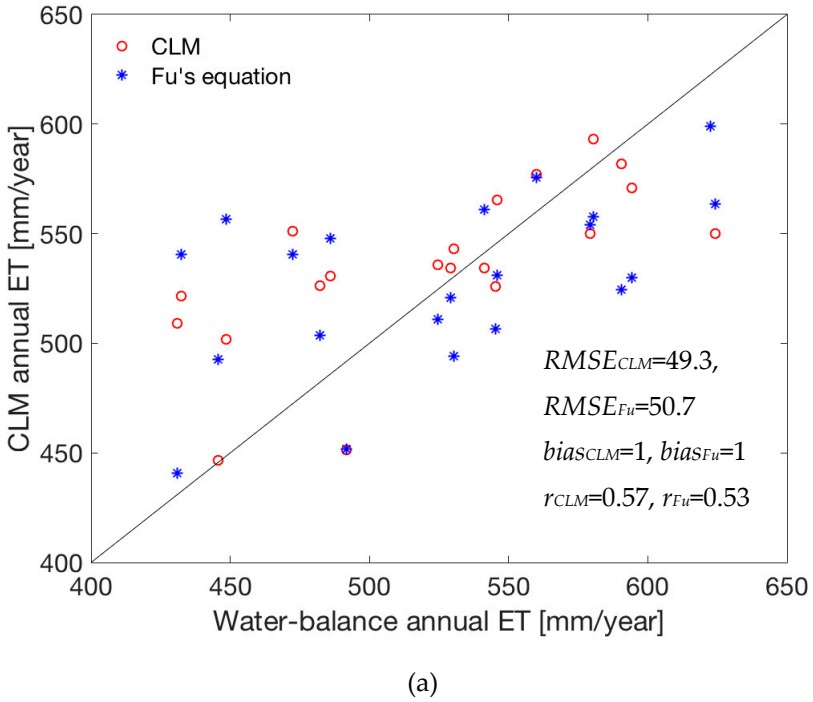

(a)

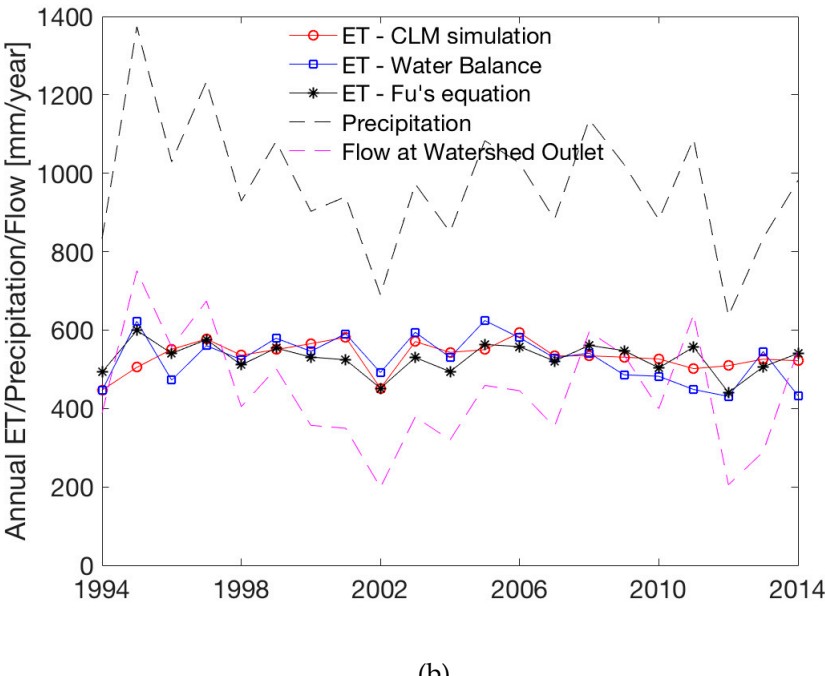

(b)

**Figure 6.** (**a**) Comparison of annual ET estimated by the CLM model and Fu's equation with the watershed-scale water balance equation. Comparison indices; RMSE, Bias and correlation coefficient (*r*) are also shown. (**b**) Annual ET estimated using the three methods, as well as estimated precipitation and river discharge at the East River Watershed during the 1994–2014 period.

Figure 6b presents the temporal variation of annual ET during the 1994–2014 water year period in comparison with that of precipitation and flow at the outlet of the watershed. The annual ET calculated using the three methods are similar and are consistently larger than the river discharge. The average annual ET over the simulation period is around 526 mm/year, which corresponds to 55% of the annual precipitation. Figure 6b also shows that the temporal variation of annual ET is

more stable than that of precipitation and river discharge. During the simulation period, while the river discharge varied significantly (from 199 to 751 mm/year), ET only fluctuated around a range of approximately 431 to 624 mm/year. There are two reasons for this fact. First, precipitation at the East river concentrates on winter season as snow. When snowmelt occurs in spring (around April), there is enough water to saturate the soil even in dry years. The remaining melting snow becomes surface flow. As a result, surface flow is highly correlated with precipitation. The second reason is that because a large portion of ET is associated with transpiration (see Figure 8), which takes water from deep soil layers via plant-water uptake. Because the temporal variation of water at deep soil layers is more stable than precipitation, ET is more temporally stable than precipitation. This will be discussed further in Sections 3.3 and 3.4. Compared to the estimation of Fu's equation and water balance equation, the temporal variation of annual ET estimated by the CLM model is more stable. It is because while the ET of a year calculated by Fu's equation and the water balance equation only depends on the precipitation of that year, the CLM model can account for the effect of precipitation of previous years on ET. For example, although precipitation in 2012 is very low, the ET estimated by the CLM model is still high because precipitation in 2011 is relatively high. This is an advantage of the model-based approach compared to the other ET calculation methods.

*3.2. Analysis of Spatiotemporal Variations of Evapotranspiration*

Figure 7 presents estimates of mean monthly ET obtained using CLM, which were averaged over the 1993–2014 period. The maps show that values of ET are lowest along the river valley and at the highest elevations. In winter, snow thickness is often greater than plant height in the river valley, limiting evaporative demand with only a small amount of sublimation occurring at the exposed canopy. At high elevations (>3900 m), ET is limited by very low air temperature. At elevations of 2950–3200 m, air temperature is higher than that at the river valley and high elevations and plant height is greater than snow depth, leading to more canopy sublimation. In summer, values of ET are still the greatest at elevations of 2950–3200 m and lowest at the high elevation locations (in June) and in the river valley (in July and August). In addition to the effect of elevation on air temperature and precipitation, vegetation at elevations of 2950–3200 m (evergreen and deciduous forests) shows high rates of summer transpiration, which is shown in Section 3.3.

Through an assessment of the monthly ET estimates, Figures 7 and 8 show that monthly ET is negligible (<14 mm/month) during the winter season (November to February) compared to the summer months. The mean monthly ET is the lowest in January and December with ET values of 8.1 and 8.6 mm/month, respectively. These values coincide with the lowest air temperatures and greatest snow depths. Monthly ET gradually increases from January to July, peaking at 100 mm/month. The total ET in summer (May to September) contributes up to 75% of annual ET, in which July alone accounts for 20%.

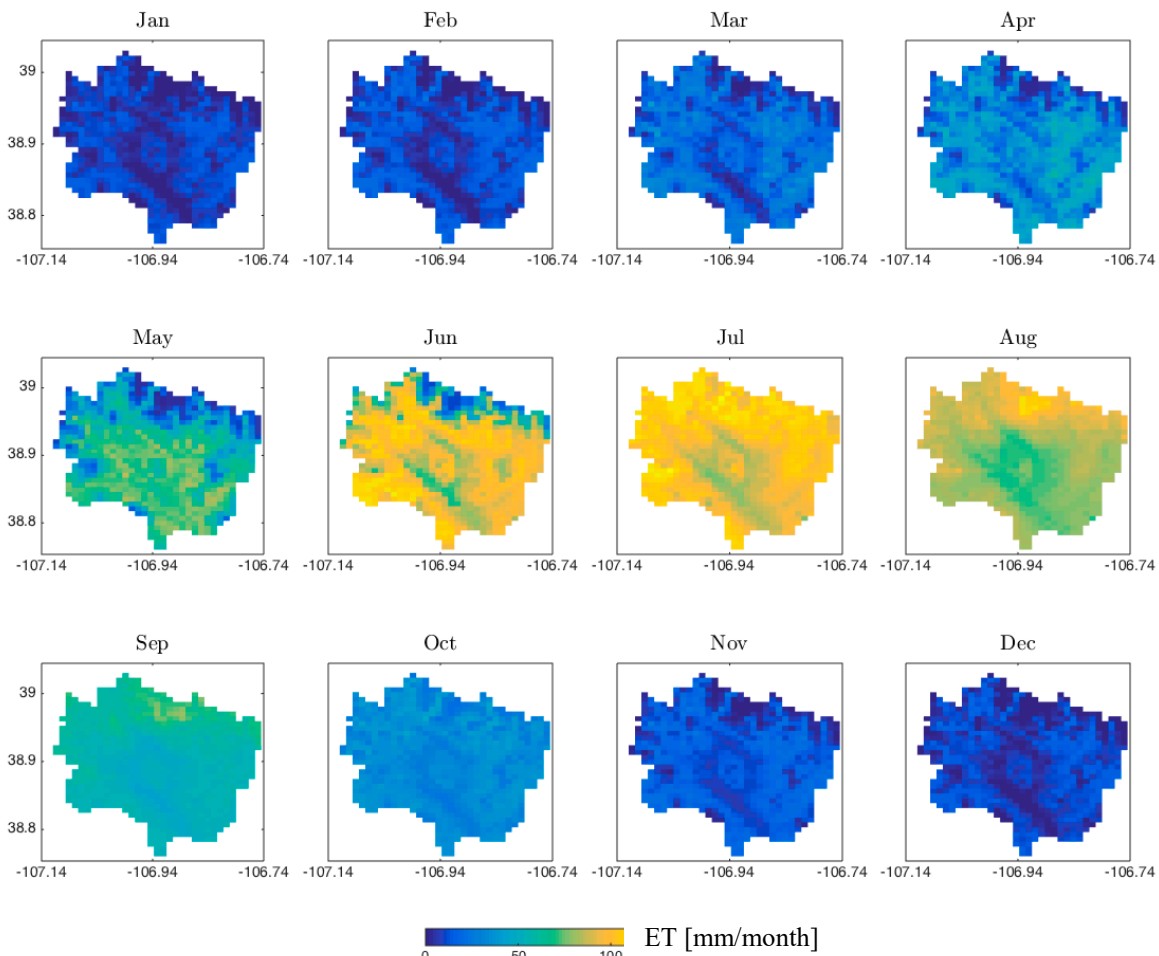

**Figure 7.** Variation of CLM-based monthly-averaged ET over time and space. The monthly-averaged ET at each grid cell was obtained by temporally averaging their corresponding values over the simulation period (1994–2014).

### 3.3. Evapotranspiration Partitioning

In addition to providing estimates of ET with high spatiotemporal resolution, the CLM also allows us to partition evapotranspiration into its components, namely, plant transpiration and soil and canopy evaporation. Here, we compared the contribution of each individual component to the total ET (Figure 8). These ET components are spatially averaged over the watershed. The figure indicates that transpiration contributes 50% of annual ET, while soil and canopy evaporation constitute 32% and 18%, respectively. However, in the winter months, the canopy evaporation is the main source of ET. This is reasonable because in winter, snow on the plant canopy is exposed to the sunlight. Therefore, it is easier for the canopy snow to sublimate than snow on the land surface. Exposure to sunlight is also the reason why transpiration is slightly larger than soil evaporation during winter (December to February) at this site. By contrast, transpiration is less than soil evaporation at the end of winter (March–April) and summer (October–November). Because leaf area index (LAI) is small during these months, it is reasonable that soil evaporation from melted snow water (in March–April) and non-frozen soil water (in October–November) is larger than transpiration. From April to September, transpiration is significantly larger than soil and canopy evaporation because in these months, vegetation develops and covers a large part of land surface. Figure 9 shows the average map of LAI from April to September to prove the effect of vegetation on total ET. As shown in Figure 9, LAI is relatively large in these months and can be up to 3.5.

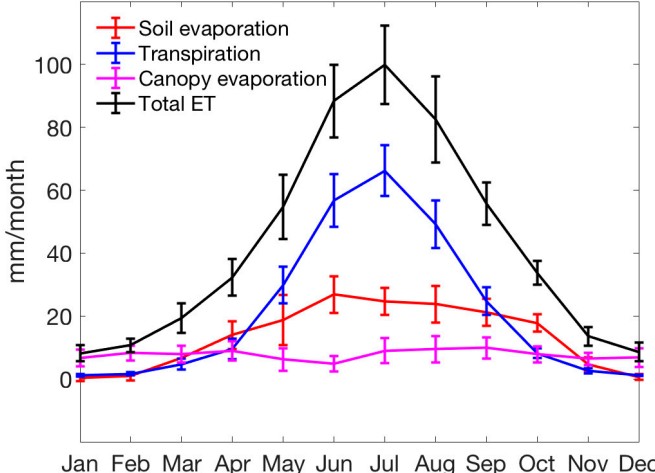

**Figure 8.** Mean monthly plant transpiration, soil and canopy evaporation and ET. The vertical bars represent the inter-annual variations of these values (± standard deviation).

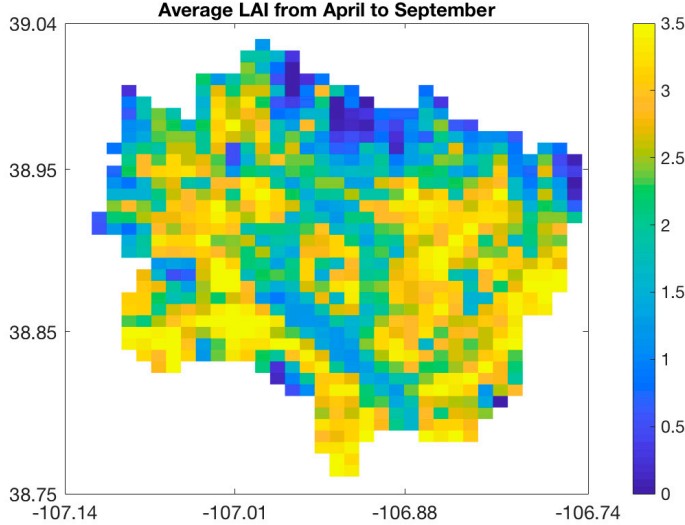

**Figure 9.** Average leaf area index (LAI) in growing season from April to September.

As for the temporal annual variation, Figure 8 also indicates that canopy evaporation is relatively stable over the year. By contrast, there are large differences in soil evaporation and transpiration in winter and summer. Soil evaporation and transpiration are negligible in winter and very large in summer. Soil evaporation increases from approximately zero in December–February to reach its maximum value (~27 mm/month) in June, while transpiration increases from zero in December-January up to 66 mm/month in July. The maximum ET of evaporation and transpiration does not occur at the same time because air temperature maximizes in July, but the soil surface is dry in this month, which limits the soil evaporation, while transpiration can take water from deep soil layers to meet the demand.

*3.4. Relationship Between Evapotranspiration and its Governing Factors*

In this study, we investigated the relationship between ET and meteorological forcing variables, surface elevation, LAI and soil texture based on the CLM simulation results. While meteorological forcing variables, surface elevation, LAI, soil texture are direct inputs of the CLM model, the surface elevation indirectly influences ET via its impacts on the spatial distribution of meteorological forcing and vegetation.

Figure 10 shows the pairwise plots of different meteorological forcing variables (precipitation, air temperature, relative humidity and shortwave radiation), topography elevation, LAI and ET. In this study, the atmospheric pressure and wind speed were not considered for analysis. The annual meteorological forcing data and ET were calculated for each grid cell and temporally averaged over the simulation period (1993–2014). Figure 9 indicates that meteorological forcing variables, which were obtained from *PRISM* and *Daymet* databases are closely correlated with elevation. Indeed, precipitation, relative humidity and shortwave radiation are positively proportional to elevation. Meanwhile, the annual mean air temperature increases from around 2.5 °C to 4 °C when elevation increases from 2600 m to around 3000 m and then decreases to −1.7 °C when elevation continues to increase to 4060 m. The near-surface air temperature (at 2 m above land surface) increases when elevation increases in the range 2600 to 3000 m. The reason for this fact is that at the lower elevation of this range, the snow layer is thicker. Because snow has a high albedo and it takes more energy to melt a thicker snow layer, the sensible heat is lower and the near-surface air temperature decreases (Pepin and Norris [29]). Similar to air temperature, ET also exhibits a concave shape in the relationship with elevation. ET increases from around 400 mm/year at an elevation of 2600 m to around 550 mm/year at an elevation of 2950–3200 m, and then reduces to 300 mm/year at the top of the watershed at an elevation of 4060 m.

Figure 10 also suggests that ET increases with increasing LAI. This is reasonable because as shown in Figure 8, the transpiration and canopy evaporation contribute to ET much more than soil evaporation. The figure also reveals that LAI and elevation are highly correlated. LAI values are greatest at elevations of 2850–3350 m and are smallest at high elevations (>3600 m). There is no clear relationship between ET and humidity and shortwave radiation. In brief, air temperature, elevation and vegetation are closely correlated to each other and are the main factors that control the spatial variation of ET.

As for the relationship between ET and soil properties, Figure 11 shows that the relatively small organic content (<4%) does not influence ET at the site. Sand and clay content show an inverse correlation with ET. Generally, at the same elevation range when the sand content is lower, the clay content is higher and ET is larger. This can be explained by the fact that soil with less sand and more clay has a higher water holding capacity. Compared to the clay content, ET is more sensitive to the variation of the sand content because sand accounts for a large portion of the soil properties. The relationship between ET and other topographical factors such as slope and aspect were also explored. However, there is no clear trend between ET and these factors.

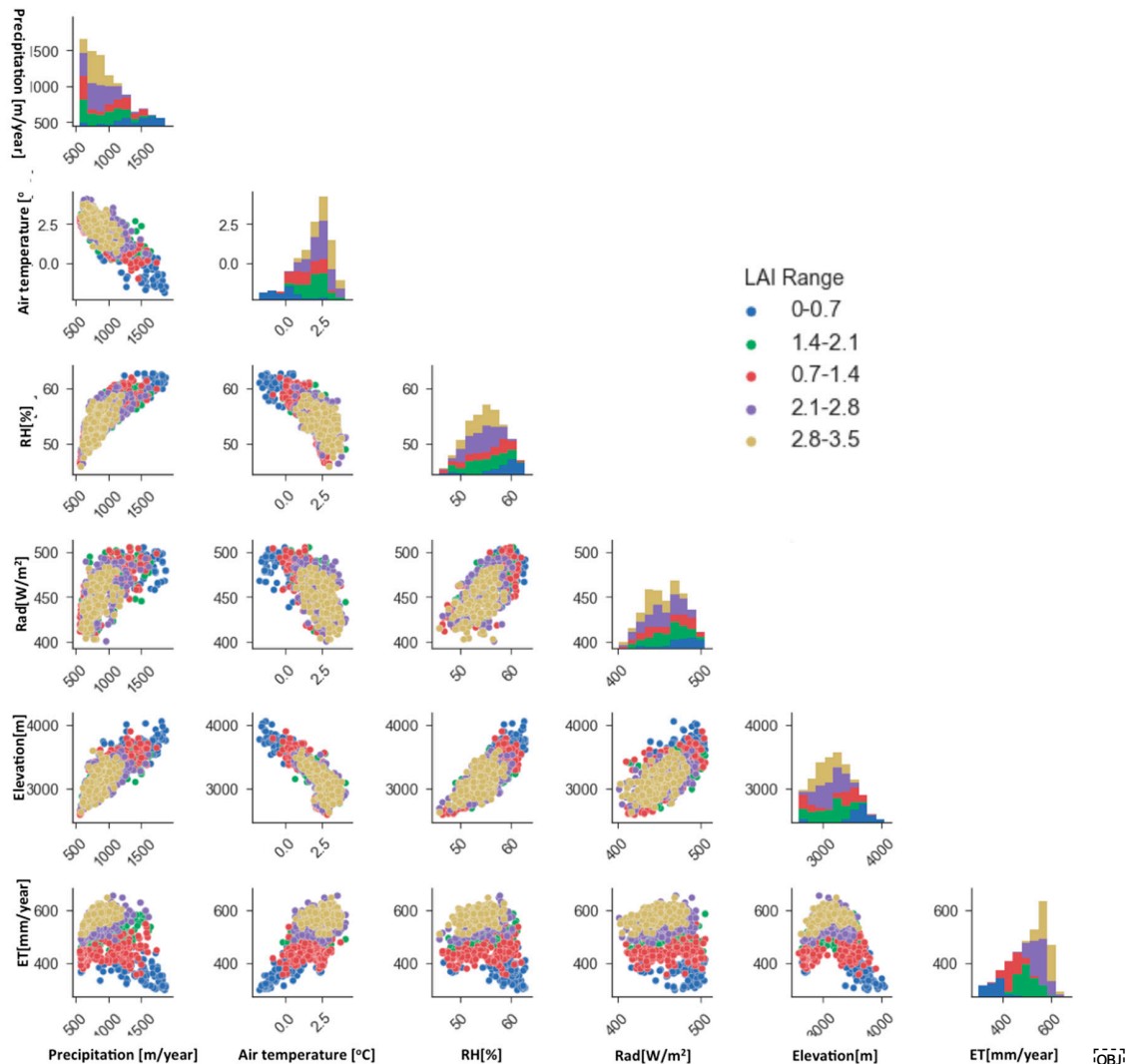

**Figure 10.** Pairwise scatter plots of meteorological forcing variables, elevation and annual ET. Different colors represent ranges of LAI values. Bar plots shows the variation of meteorological forcing variables, elevation and annual ET.

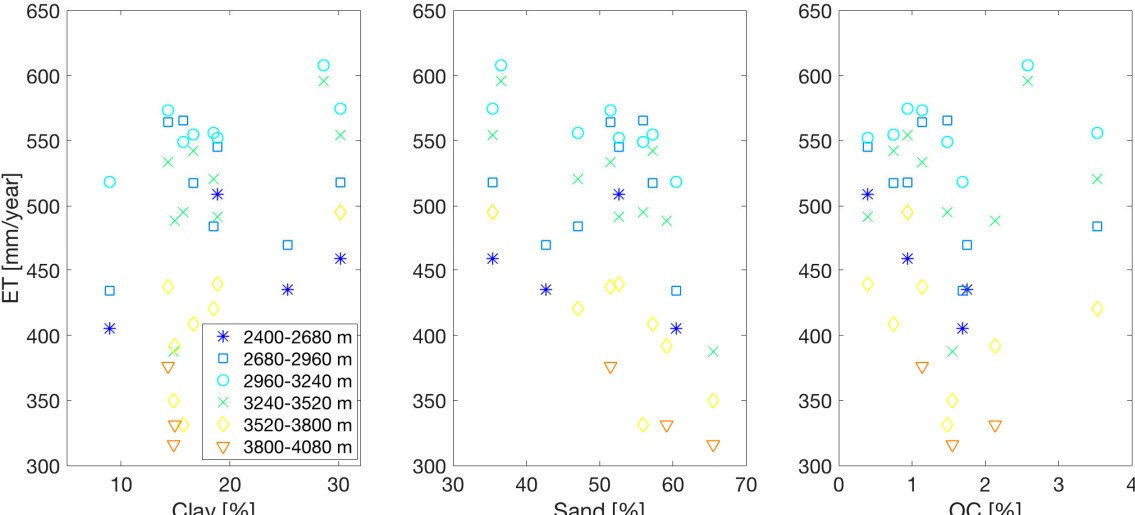

**Figure 11.** Scatter plots of clay, sand and organic carbon content and ET. Different symbols represent the different elevation ranges.

## 4. Conclusions

This study assessed the spatiotemporal variations of ET and analyzed the relationship between ET and its controlling factors at the mountainous East River watershed, Colorado, U.S. The CLM model was used to estimate the ET with a high spatiotemporal resolution. The comparison of the CLM model output with the results obtained from the watershed-scale water balance equation and the Fu equation showed that the ET estimated by the three methods was in a good agreement with the bias criterion of approximately 1.

As for the temporal variation of ET, the annual ET is much more temporally stable than the river discharge. The temporal analysis of monthly ET shows that the summer period from May to September contributes up to 75% of annual ET, of which ET in July accounts for 20%. In winter, ET is low (<14 mm/month) and mostly comes from the canopy evaporation. As for the spatial variation, ET along the river valley (<2750 m) and at high elevations (>3900 m) is much smaller than that at an elevation range of 2950—3200 m because at this elevation range the air temperature is highest and the LAI is largest, which is convenient for ET.

This is one of the first studies that explored all of the factors that potentially control the spatiotemporal variations of ET. We found that at the Upper East River Watershed, elevation, vegetation type and air temperature are the main factors that appear to influence the spatial variation of ET. However, these quantities are closely correlated with each other. Finally, the soil with more clay slightly increases ET because it increases the soil water hold capacity and decreases the hydraulic conductivity, and therefore, reduces the amount of water lost to groundwater drainage. Because the CLM model calculates the soil's physical parameters only from the sand, clay and organic carbon content, this study did not account for the effect of other soil types such as silt.

This study provides a promising approach to the estimation of ET and assesses its spatiotemporal variations at the mountainous watershed scale with high spatiotemporal resolution. The analysis of the correlation between ET and its controlling factors also provides important knowledge to determine the hydrological hot spots and to scale up from the local to watershed scale. Future research should include combining our physically-based model with a data-driven model to take advantage of the benefits of both models for improving both the accuracy and computational time of estimating ET.

**Author Contributions:** Conceptualization, A.P.T., B.F. and B.D.; Data curation, A.P.T. and J.R.; Formal analysis, J.R.; Funding acquisition, B.D. and S.S.H.; Investigation, A.P.T.; Methodology, A.P.T., J.R. and B.F.; Supervision, B.D. and S.S.H.; Validation, A.P.T.; Visualization, A.P.T.; Writing—original draft, A.P.T.; Writing—review & editing, A.P.T., J.R., B.F., B.D. and S.S.H.

**Funding:** This research was funded by the U.S. Department of Energy, Office of Science, Office of Biological and Environmental Research under Award Number DE-AC02-05CH11231.

**Acknowledgments:** This material is based upon work supported as part of the Watershed Function Scientific Focus Area funded by the U.S. Department of Energy, Office of Science, Office of Biological and Environmental Research under Award Number DE-AC02-05CH11231.

**Conflicts of Interest:** The authors declare no conflict of interest.

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
