# Peer review of "Assessment of Spatiotemporal Variability of Evapotranspiration and Its Governing Factors in a Mountainous Watershed"

_water, doi:10.3390/w11020243_

Round 1
Reviewer 1 Report
The comments and suggestions for improvement are listed in the attached document "Comments_423524.pdf". Particularly the presentation of the results in the illustrations should be improved. Some clarifications regarding content are also important. The bibliography is not yet consistent and accurate.

Author Response
We would like to thank the first reviewer for his/her efforts to review our study, which definitely helped to significantly improve the quality of our paper. Below are our answers to the reviewer's comments and suggestions.
22: “coarser soils”: “coarse textured soils” would be correct.
23: “southwest-facing aspects”: “southwest-facing slopes” would be better. The statement on influence of slope should be proven, see below.
47: aa instead of a. As a result of what? The agumentation is not clear. 64 f: fine (coarse) texture soils instead of fine (coarse) textured soil 75: Thirdly instead of secondly
119: (8) instead of (4).
111-122: inconsistent type size of equations
Reply: Those typos errors were corrected in the revised manuscript
179: “i.e., atmospheric pressure is larger for higher elevation.”? I don’t think so.
Reply: We changed to ‘atmospheric pressure is smaller for higher elevation.’
181: labeling of subfigures with letters is very dominant, could be at an upper corner. This is also true for the other figures with sub figures.
Reply: New figures with smaller titles were added
195: “900” unmotivated bold face 204: needle evergreen occurs twice.
212: replace hydrological-thermal by soil physical
Reply: We corrected those suggestions in the revised manuscript
219: the labeling of sub figures is too prominent and partly redundant, letters are sufficient
Reply: We kept the name of soil types so that readers know the spatial distribution of these soil types, but we reduce the size of texts
235: What means approximately? Quantify it.
Reply: We remove word ‘approximately’. The bias values for two methods are greater than 0.99.
247, 250: river discharge is may be more adequate than river flow
Repy: We used river discharge instead of river flow
291: huge letters and numbers of figure 8 compared to other figures and text 318: which figure? Figure 9 indicates …
333: “soil texture” should be replaced by “soil properties” because organic matter is not part of texture.
338: “because sand constitutes a large portion in soil texture.” I don’t agree with the interpretation and the formulation is not clear. Why didn’t you integrate the silt content in your analysis?
Reply: In CLM model, only three soil properties are used to calculate physical parameters including sand, clay and soil organic carbon content. Silt content are not considered so we have not considered it in our analysis.
339 ff: I doubt, whether the described effects can be derived from the data shown in figure 11. Are the trends statistically proven? Moreover, only the slope is considered although the aspect has a clear effect on ET, connected with the slope.
Reply: We agree with the reviewers. Due to low statistical correlation between ET and slope, we removed the part that discuss the relationship between ET and slope.
358: axis labeling not aligned
Reply: This was corrected in the revised manuscript
360: significance of bar plots, at the position of autocorrelation
Reply: Bar plots shows the variation of meteorological forcing variables, elevation and annual ET
362: To improve the readability of the figure 10 the point signatures and colors should have logical order (e.g. rainbow colours, …).
Reply: Rainbow colors were used for new figure in the revised manuscript (Figure 11)
383 f: The air temperature is not highest in middle elevation see fig. 9!
Reply: ‘Middle elevation’ was removed
390: As mentioned before I think slope and aspect should be analyzed together.
Reply: Due to low statistical correlation between ET and slope as well as ET and aspect, we did not consider those relationships in our revised manuscript.
391: As already mentioned silt content should be considered when investigating water holding capacity
Reply: we added a sentence to mention that this study has not considered the effect of silt content for our analysis because the CLM model does not include this content in its formulae that calculate physical properties from sand, clay and organic carbon content.
403 ff: the section references is formally not consistent, many typos (with oder without “pp.”, doi- adress, …).
Reply: We corrected the references and added DOI address to each paper.
Reviewer 2 Report
General comments:
The manuscript presents the spatiotemporal variability of evapotranspiration (ET) and its governing factors in a mountainous watershed in the Upper Colorado River Basin. The study compared the ET results from Community Land Model (CLM) with estimated ET from watershed-scale water balance equation. The results presented in the manuscript is interesting and is suitable for the aims and scope of the journal. However, the manuscript needs improvement before publication. Authors are also requested to revisit the instruction for authors and update the manuscript (citation style for example) as per the journal’s requirement. The major concerns are detailed below.
Comment 1: RMSE formula is not presented well in Page 8 - Line 230. The unit must be mm/year. The RMSE of 2.43 x 103 and 2.57 x 103 mm2 (Line 233) does not make sense when annual ET is less than 650 mm/year (Figure 6a). Please update the formula and explanation in the text accordingly.
Comment 2: Add a table for statistical comparison (R2, bias, RMSE) for CLM and Fu’s equation with water balance estimation or add R2, bias, and RMSE in Figure 6a. In Figure 6b, put precipitation in the secondary axis.
Comment 3: The spatiotemporal variability of ET is discussed based on the CLM simulations. The annual ET from CLM simulation is relatively flat (low variation) compared to Fu’s equation and water balance equation (Figure 6b). For example, from year 2010 to 2014, ET remains almost similar. In contrast, precipitation and flow show high variability during the same time period. What might be the reason for CLM ET estimation not being sensitive to precipitation variation? Also, the annual ET from CLM for 2012 (possible drought year- lowest rainfall during the study years) is almost similar or higher compared to 2011, while it is lower in 2012 compared to 2011 from other methods (Fu’s equation and water balance). Please add main equation(s) for estimating ET within CLM in section 2.1.1.
Comment 4: Add LAI maps for the months with higher ET rates (May-September) as the study shows 50% of ET is contributed by transpiration and LAI (vegetation) as one of the primary drivers of ET in CLM. The LAI maps may help to strengthen the finding of the study for ET variation with respect to elevation. Also, discuss the results with previous studies.
Specific comments:
Line 43: …..include the Community Land Model?
Line 47: As a? result, ………
Line 49: ….also intensively studied?
Line 123: ‘As a result, we cannot estimate monthly ET for this case study.’ Clarify this sentence…
Line 303-306: this paragraph is not providing any additional result or scientific value, this paragraph can be removed
Line 321: ‘…..mean air temperature increase from around 2.5 oC to 4 oC when elevation increases from 2600 m to around 2900-3100 m and then decrease to -1.7 oC when elevation increase to 4060 m….’ . Please add an explanation. Usually, air temperature is expected to decrease with the increase in elevation.
Author Response
The manuscript presents the spatiotemporal variability of evapotranspiration (ET) and its governing factors in a mountainous watershed in the Upper Colorado River Basin. The study compared the ET results from Community Land Model (CLM) with estimated ET from watershed-scale water balance equation. The results presented in the manuscript is interesting and is suitable for the aims and scope of the journal. However, the manuscript needs improvement before publication. Authors are also requested to revisit the instruction for authors and update the manuscript (citation style for example) as per the journal’s requirement. The major concerns are detailed below.
Comment 1: RMSE formula is not presented well in Page 8 - Line 230. The unit must be mm/year. The RMSE of 2.43 x 103 and 2.57 x 103 mm2 (Line 233) does not make sense when annual ET is less than 650 mm/year (Figure 6a). Please update the formula and explanation in the text accordingly.
Reply: RMSE formula and its values were updated in lines of the revised manuscript
Comment 2: Add a table for statistical comparison (R2, bias, RMSE) for CLM and Fu’s equation with water balance estimation or add R2, bias, and RMSE in Figure 6a. In Figure 6b, put precipitation in the secondary axis.
Reply: correlation coeficient, bias and RMSE were added to Figure 6a.
In Figure 6b, Precipitation and ET have the same unit so we will keep them at the same axis for comparison.
Comment 3: The spatiotemporal variability of ET is discussed based on the CLM simulations. The annual ET from CLM simulation is relatively flat (low variation) compared to Fu’s equation and water balance equation (Figure 6b). For example, from year 2010 to 2014, ET remains almost similar. In contrast, precipitation and flow show high variability during the same time period. What might be the reason for CLM ET estimation not being sensitive to precipitation variation? Also, the annual ET from CLM for 2012 (possible drought year- lowest rainfall during the study years) is almost similar or higher compared to 2011, while it is lower in 2012 compared to 2011 from other methods (Fu’s equation and water balance). Please add main equation(s) for estimating ET within CLM in section 2.1.1.
Reply: Following explanations were added to the revised manuscript:
Figure 6b also shows that the temporal variation of annual ET is more stable than that of precipitation and river discharge. During the simulation period, while the river discharge varied significantly (from 199 to 751 mm/year), ET only fluctuated around a range of approximately 431 to 624 mm/year. There are two reasons for this fact. First, precipitation at the East river concentrates on winter season as snow. When snowmelt occurs in spring (around April), there is enough water for saturating soil even in dry years. The remaining melting snow becomes surface flow. As a result, surface flow highly correlates with precipitation. The second reason is that because a large portion of ET is associated with transpiration (see Figure 8), which takes water from deep soil layers via plant-water uptake. Because the temporal variation of water at deep soil layers is more stable than precipitation, ET is more temporally stable than precipitation. This will be discussed further in sections 3.3 and 3.4. Compared to the estimation of Fu’s equation and water balance equation, the temporal variation of annual ET estimated by the CLM model is more stable. It is because while the ET of a year calculated by Fu’s equation and the water balance equation only depends on the precipitation of that year, the CLM model can account for the effect of precipitation of previous years on ET. For example, although precipitation in 2012 is very low but ET estimated by the CLM model is still high because precipitation in 2011 is relatively high. This is an advantage of the model-based approach compared to the other ET calculation methods.
Comment 4: Add LAI maps for the months with higher ET rates (May-September) as the study shows 50% of ET is contributed by transpiration and LAI (vegetation) as one of the primary drivers of ET in CLM. The LAI maps may help to strengthen the finding of the study for ET variation with respect to elevation. Also, discuss the results with previous studies.
Reply: the average LAI map in the April-September period was added to the revised manuscript (Figure 9)
Specific comments:
Line 43: …..include the Community Land Model?
Reply: We changed to “Community Land Model” in the revised manuscript
Line 47: As a? result, ………
Reply: This typo error was corrected in the revised manuscript
Line 49: ….also intensively studied?
Reply: This typo error was corrected in the revised manuscript
Line 123: ‘As a result, we cannot estimate monthly ET for this case study.’ Clarify this sentence…
Reply: We changed to ‘As a result, we could not estimate monthly ET for this case study
Line 303-306: this paragraph is not providing any additional result or scientific value, this paragraph can be removed
Reply: The paragraph was removed
Line 321: ‘…..mean air temperature increase from around 2.5 oC to 4 oC when elevation increases from 2600 m to around 2900-3100 m and then decrease to -1.7 oC when elevation increase to 4060 m….’ . Please add an explanation. Usually, air temperature is expected to decrease with the increase in elevation.
Reply: The following explanation was added to the revised manuscript:
The near-surface air temperature (at 2 m above land surface) increases when elevation increases in a range 2600 to 3000 m. The reason for this fact is that at the lower elevation of this range, snow layer is thicker. Because snow has high albedo and it takes more energy to melt thicker snow layer, the sensible heat is lower and the near-surface air temperature decreases (Pepin and Norris, 2005).
Round 2
Reviewer 2 Report
The current version of the manuscript addressed the issues it had in the previous version. Recommended for publication. (Note: may need to double check the citation style)